# Mucoepidermoid Carcinoma of the Uterine Cervix—Single-Center Study Over a 10-Year Period

**DOI:** 10.3390/medicina56010037

**Published:** 2020-01-18

**Authors:** Angel Yordanov, Martin Karamanliev, Latchezar Tantchev, Assia Konsoulova, Strahil Strashilov, Mariela Vasileva-Slaveva

**Affiliations:** 1Department of Gynecologic Oncology, Medical University Pleven, 5800 Pleven, Bulgaria; 2Department of Surgical Oncology, Medical University Pleven, 5800 Pleven, Bulgaria; martinkaramanliev@gmail.com; 3Obstetrics and Gynecology Clinic, Acibadem City Clinic Hospital “Tokuda”, PC 1000 Sofia City, Bulgaria; 4Department of Medical Oncology, Complex Oncological Center Burgas, 8000 Burgas, Bulgaria; dr.konsoulova@gmail.com; 5Department of plastic and reconstructive surgery, MU-Pleven, 5800 Pleven, Bulgaria; dr.strashilov@gmail.com; 6EXTRO-Lab, Department of Therapeutic Radiology and Oncology, Medical University of Innsbruck, 6020 Innsbruck, Austria; sscvasileva@gmail.com; 7Tyrolean Cancer Research Institute, 6020 Innsbruck, Austria; 8EORTC Pathobiology Group, 1200 Brussels, Belgium

**Keywords:** mucoepidermoid cervical carcinoma, adenosquamous carcinoma, survival rate, lymph node involvment

## Abstract

*Background and objectives:* Adenosquamous cancer of the uterine cervix is a rare type of cervical cancer with both malignant squamous and glandular components. A very rare subtype is mucoepidermoid carcinoma (MEC), which was first described as a salivary gland tumor. It has been described as having the appearance of a squamous cell carcinoma without glandular formation and contains intracellular mucin. The postoperative evolution of this tumor and the potentially poorer prognosis may indicate an intensification of the follow-up. The objective of our study was to analyze the frequency of mucoepidermoid carcinoma in hospitalized women with cervical cancer, clinical characteristics and prognosis. *Material and Methods:* A retrospective study of all cases of mucoepidermoid carcinoma of the cervix at Department of Gynecologic Oncology, University Hospital—Pleven, Pleven Bulgaria between 1 January 2007 and 31 December 2016 was performed. All patients were followed-up till December 2019. We analyzed certain clinical characteristics of the patients; calculated the frequency of mucoepidermoid carcinoma of the cervix from all patients with stage I cervical cancer; and looked at the overall survival rate, correlation between overall survival, lymph node status and the size of the tumor. *Results:* The frequency of MEC was 1.12% of all patients with stage I cervical cancer in this study. The median age of the patients with MEC was 46.7 years (range 38–62). Four patients (57.1%) were staged as FIGO IB1, and three patients (42.8%) were FIGO IB2. The size of the primary tumor was <2 cm in 2 patients (28.57%), 2–4 cm in 2 patients (28.57%) and >4 cm in 3 patients (42.8%). Metastatic lymph nodes were found in two patients (28.57%), and nonmetastatic lymph nodes were found in five patients (71.43%). There were two (28.57%) disease-related deaths during the study period. The five-year observed survival in the MEC group was 85.7% and in the other subtypes of adenosquamous cancer group was 78.3%. *Conclusions:* MEC of the uterine cervix is a rare entity diagnosis. As a mucin-producing tumor, it is frequently regarded as a subtype with worse clinical behavior and patients’ outcomes. Nevertheless, our data did not confirm this prognosis. New molecular markers and better stratification are needed for better selection of patients with CC, which may benefit more from additional treatment and new target therapies.

## 1. Introduction

Adenosquamous cancer (ASC) of the uterine cervix is a rare type of cervical cancer (CC), and its frequency is between 3% and 10% [1,2,3]. This tumor is a biphasic variant of CC featuring both malignant squamous and glandular components [4]. ASC of the cervix must be distinguished from large cell nonkeratinising squamous cell carcinomas and endometrioid adenocarcinomas with squamous metaplasia in which the squamous component is benign [3]. Still, it carries genetic and protein signatures of both squamous cell carcinoma (SSC) and adenocarcinoma AC [5]. For diagnosis, it is necessary that both squamous and glandular elements are recognized on routine H&E sections without applying special stains or immunohistochemistry [4]. ASC has a prognosis than SSC [1] and a trend for a worse prognosis than adenocarcinoma AC, which does not reach significance [6] and therefore is not placed in the independent prognostic category. It is usually poorly differentiated/undifferentiated, and is more frequently diagnosed in stage I [1].

ASC of the cervix is extremely rare and when it displays three cells types: epidermoid, mucin-producing and intermediate, it is named mucoepidermoid carcinoma (MEC) [7]. This subtype of adenocarcinoma was first described as a salivary gland tumor by Stewart et al. [8] and occurs in about 30% of all malignant tumors of the major or minor salivary gland [9]. It can also occur in other localizations such as lung and esophagus [10], and it is extremely rarely observed in uterine cervi [11]. It has been described as having the appearance of a squamous cell carcinoma without glandular formation and contains intracellular mucin [11]. There are authors suggesting that the postoperative evolution of this tumor and the potentially poorer prognosis may indicate an intensification of the follow-up [12]. We aimed to investigate the prevalence and the prognosis of MEC among the patients treated for a 10-year period of time in our institution.

## 2. Material and Methods

An ethical committee approval (number 414-KEHИД/31.03.2016) was obtained to perform a retrospective study of all patients who operated for cervical cancer at Clinic of Oncogynaecology, University Hospital—Pleven, Bulgaria for a 10-year period between 1 January 2007 and 31 December 2016. All mucoepidermoid histologies were centrally reviewed in order to reconfirm the diagnosis. The information regarding demographic characteristics consisted of age at diagnosis, date and type of surgery, and clinical staging according to TNM and FIGO classification; furthermore, postoperative management, postoperative staging, lymph node metastatic and follow-up were obtained from medical history records. Patients were followed up until 1 December 2019. The standard followed up included clinical examination and blood workup every 3 months during the first 2 years and then annually, as well as annual whole-body CT. We compared the investigated clinicopathological characteristics: age at diagnosis, type of surgery, clinical and pathological staging according to TNM and FIGO classification and lymph node status among the patients with MEC and other adenosquamous carcinoma subtypes. We investigated the OS among these two groups. Statistical analysis was performed on SPSS software.

## 3. Results

During the study period, 624 patients with cervical cancer FIGO I stage were operated on in our clinic. Thirty one patients (4.97%) were diagnosed with adenosquamous carcinoma. Seven of them (22.58% from the adenosquamous carcinoma and 1.12% of all) had histology for MEC, and 24 patients were with other subtypes of adenosquamous carcinoma (control group).

All patients had initially histologically confirmed diagnosis prior to surgery via biopsy or dilation and curettage, and had undergone radical hysterectomy with total pelvic lymph node dissection, followed by postoperative radiotherapy (Figure 1).

The median age of patients with MEC was 46.7 years (range 38–62). Four patients (57.1%) were staged as FIGO IB1, and three patients (42.8%) were FIGO IB2 stage. Patient’s characteristics are shown in Table 1; Table 2. There were two (28.57%) disease-related deaths during the study period (Table 1). It is ntable that the tumor in those diseased patients was >4 cm and they both had lymph node metastases. The five-year observed survival was higher in the MEC group, but the differences were not statistically significant (p0892, Figure 2).

The mean age at diagnosis is almost the same in patients with adenosquamous carcinoma and MEC. Despite the small number of patients, we observed a trend for more lymph node metastasis in patients with pure adenosquamous cancer.

## 4. Discussion

Cervical cancer is the 4th most common cause of cancer-related death in the world [13]. Treatment for early stage CC includes surgery (such as cervical conization, total simple hysterectomy, or radical hysterectomy) with bilateral lymph node dissection and adjuvant therapy according to the presence of combination of risk factors such as lymph node involvement, and tumor differentiation, but independently of the histological type [14]. MEC of the uterine cervix is a rare subtype of ASC. Its prevalence was very low also among the patients treated in our hospital. There are specific criteria for this diagnosis: prevalence of epidermoid cells, scattered intermediate cells, and cells containing intracellular mucin with no glandular formation. Additionally, MEC tumors show specific genetic rearrangements [15], which could be used in the future development of target therapy [5]. They also show specific immunohistochemical profile (antiCEA stains are positive and antivimentin are negative in cervical MEC carcinoma, whereas in endometrial MEC it is the opposite: antiCEA is negative and antivimentin is positive) [16], which distinguishes them from other cervical cancers and endometrial MEC. It was interesting then for us to investigate if they have different metastatic pattern and survival outcomes.

MECs are mucin-producing tumors, and mucin production is considered to be a worse prognostic factor for those tumors [12]. These tumors were considered to have greater potential for metastatic spread to the regional lymph nodes [17,18,19]. Lymph node metastasis was stated to be an independent prognostic factor for overall survival [11,20].

In the literature, MEC of the uterine cervix is very rarely described [11,21,22] and its frequency is not well reported [21]. Recent data showed that tumors, diagnosed as MEC according to morphology, carry genetic rearrangements and mutations in the same genes, as MECs of the salivary glands [22]. These lead to the suggestion that MEC is a separate disease rather than a subtype of the cervical adenosquamous carcinoma [23], and that it might have more aggressive clinical behavior [11]. In a case series of 12 women, followed for 2–15 years, three of them died within 14 months and they all had lymphogenic and distant metastases. The incidence of lymph node metastases was reported in 33% of the cases.

Kim et al. [22] reported a case of MEC of cervix staged as 1B1. They did not prefer adjuvant therapy; however, the tumor recurred in the fourth month, and the patient died 19 months after surgery.

Different clinical behavior of cervical MEC was also reported by other groups, suggesting different clinical evolution from adenosquamous cancer with higher rates of lymph node metastases as well as comparable radiosensitivity [24].

Despite these differences between MEC and adenosquamous cancer, they are still classified as common conditions and their treatment is similar. Data about the treatment of MEC are scarce, and no management guidelines have been developed. Although data in the literature suggest more aggressive clinical behavior of MEC of the cervix, our data do not confirm this. Data from this study indicate more lymph node metastasis in other ASC sub-types than in the MEC subtype. Our survival data do not show a significant difference in overall survival between MEC and the other subtypes of adenosquamous cancer. The five-year observed survival was higher in MEC group, but not statistically significant.

## 5. Conclusions

MEC of the uterine cervix is a rare diagnosis. As a mucin-producing tumor, it is frequently regarded as a subtype with worse clinical behavior and patients’ outcomes. Nevertheless, our data did not confirm its poorer prognosis. Still, new molecular markers and better stratification are needed for better selection of patients with CC, who may benefit more from additional treatment and new targeted therapies.

## Figures and Tables

**Figure 1 medicina-56-00037-f001:**
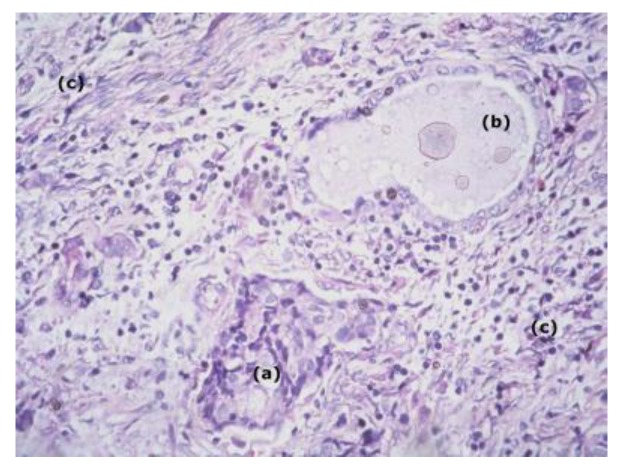
Microscopic finding haematoxylin and eosin stain 200x. (**a**) epidermoid cells; (**b**) mucin-producing cells; (**c**) inmtermediate cells.

**Figure 2 medicina-56-00037-f002:**
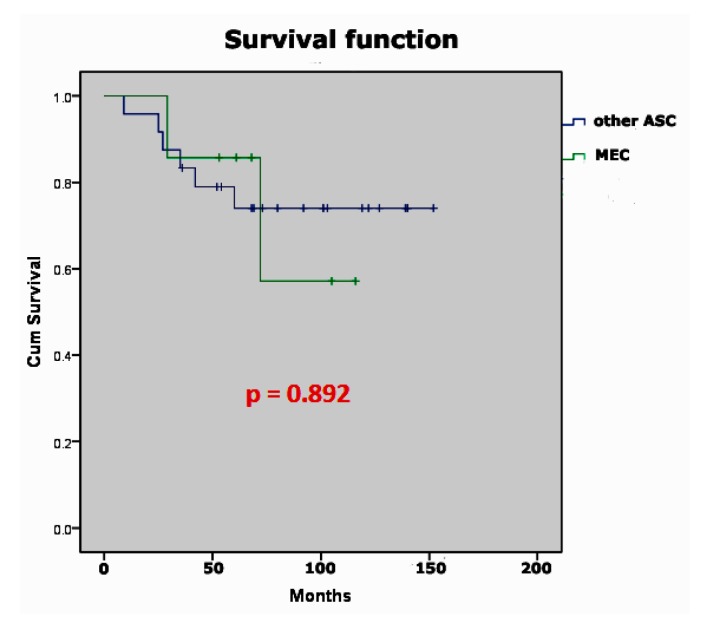
Overall survival of mucoepidermoid cancer of the cervix and other subtypes of adenosquamous cancer of the cervix.

**Table 1 medicina-56-00037-t001:** Patient’s characteristics according to tumor histology.

Patient’s Characteristics	Other Adenosquamous	MEC
Mean age at diagnosis	48	47
Size of primary tumor <2, n (%)	11 (45.8%)	2 (28.6%)
Size of primary tumor 2–4, n (%)	3 (12.5%)	2 (28.6%)
Size of primary tumor >4, n (%)	10 (41.7%)	3 (42.9%)
positive lymph nodes (N+)	8 (33.3%)	2 (28.6%)
Median OS	70	76.5
Five years observed survival	78.3%	85.7%
Total	24	7

**Table 2 medicina-56-00037-t002:** Clinical and pathological characteristics of the patients with mucoepidermoid carcinoma (MEC), outcome and survival.

Case	Age (Years)	Pathological Stage	Size of Primary Tumor (cm)	Outcome	Survival (Months)
	41	pT1b2pN1M0	>4	Dead	72
2	42	pT1b1pN0M0	<2	Alive	105
3	38	pT1b2pN1M0	>4	Dead	29
4	41	pT1b1pN0M0	B/n 2 and 4	Alive	68
5	55	pT1b1pN0M0	<2	Alive	61
6	48	pT1b2pN0M0	>4	Alive	116
7	62	pT1b1pN0M0	B/n 2 and 4	Alive	53

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
