# Peer review of "Mucoepidermoid Carcinoma of the Uterine Cervix—Single-Center Study Over a 10-Year Period"

_medicina, 2020, doi:10.3390/medicina56010037_

Round 1
Reviewer 1 Report
Definitions and illustrations not clear.
Needs significant language corrections for greater clarity.
Details are in notes in uploaded reviewed manuscript.

Reviewer 2 Report
Angel Yordanov and colleagues investigated Mucoepidermoid carcinoma of the uterine cervix - Single-center study over a 10-year period. The postoperative evolution of this tumor and the potentially poorer prognosis may indicate an intensification of the follow-up. The objective of this study was to analyse the frequency of mucoepidermoid carcinoma in hospitalized women with cervical cancer, clinical characteristics and prognosis. This data didn’t confirm its worse prognosis than previous reports. This report has a small number of cases compared to the previous reports. However, this paper is the greatest contribution to rare tumor of uterine cervix.
Minor comments:
Mucoepidermoid carcinoma of the uterine cervix is very rare tumor and pathological diagnosis is important.
The authors should include criteria for pathological diagnosis for Mucoepidermoid carcinoma of the uterine cervix in “Material and Methods”. If necessary, the author should add typical photographs of histopathology (HE stains and immunohistochemical stains of tumor).
